# Demographic characteristics, long-term health conditions and healthcare experiences of 6333 trans and non-binary adults in England: nationally representative evidence from the 2021 GP Patient Survey

Catherine L Saunders [ID],[1] Alison Berner [ID],[2,3] Jenny Lund,[1] Amy M Mason,[4] Tash Oakes-Monger,[5] Meg Roberts,[6] Jack Smith,[7] Robbie Duschinsky [ID] [1]

For numbered affiliations see end of article.

**Correspondence to**
Dr Catherine L Saunders;
ks659@medschl.cam.ac.uk

## ABSTRACT

**Objective** In order to address the lack of data on the health and healthcare needs of trans and non-binary adults, NHS England includes questions asking about both gender and trans status in its surveys to support quality improvement programmes.

We used self-reported data from the GP Patient Survey to answer the research question: what are the demographic characteristics, health conditions and healthcare experiences of trans and non-binary adults in England?

**Design/setting** Nationally representative, population-based cross-sectional survey in England with survey data collection from January to March 2021.

**Participants** 840 691 survey respondents including 6333 trans and non-binary adults.

**Outcomes** We calculated weighted descriptive statistics, and using logistic regression explored 15 long-term physical and mental health conditions, and 18 patient experience items, covering overall experience, access, communication and continuity.

**Results** Trans and non-binary adults were younger, more likely to be from Asian, black, mixed or other ethnic groups and more likely to live in more deprived parts of the country. Age-specific patterns of long-term conditions were broadly similar among trans and non-binary adults compared with all other survey respondents, with some variation by condition. Overall, inequalities in long-term health conditions were largest for autism: OR (95% CI), 5.8 (5.0 to 6.6), dementia: 3.1 (2.5 to 3.9), learning disabilities: 2.8 (2.4 to 3.2) and mental health: 2.0 (1.9 to 2.2), with variation by age. In healthcare experience, disparities are much greater for interpersonal communication (OR for reporting a positive experience, range 0.4 to 0.7 across items) than access (OR range 0.8 to 1.2). Additionally, trans and non-binary adults report much higher preference for continuity 1.7 (1.6 to 1.8), with no evidence of any differences in being able to see or speak to a preferred general practitioner.

**Conclusion** This research adds up to date evidence about population demographics, health and healthcare needs to

## STRENGTHS AND LIMITATIONS OF THIS STUDY

⇒ Information about the demographics, health and healthcare needs of people who are trans and non-binary is still emerging; what evidence there is suggests disparities are high.

⇒ This study provides nationally representative population-based evidence on long-term conditions and healthcare experiences needed to inform healthcare planning and quality improvement for trans and non-binary adults, particularly in primary care.

⇒ Although this work was not fully coproduced, the protocol was developed in collaboration with a trans, non-binary and queer patient and public involvement panel, who were involved throughout the whole study.

support healthcare improvement for trans and non-binary adults.

## INTRODUCTION

Primary care supports populations across the life course, responding to health and healthcare needs that change over time, and with age, and developments in clinical practice. Care for people who are trans and non-binary is one area where GPs are providing health services, while information about the demographics, health and healthcare needs of the population is still emerging.[1–6]

Addressing gaps in health access and outcomes experienced by minority groups including those with protected characteristics—those characteristics for which it is against the law in the UK to discriminate against someone and which include gender reassignment—is central to the priorities of

the National Health Service (NHS) long-term plan.[7][8] Primary care is important for these efforts to prevent ill health and address health inequalities. Clinical guidelines for best practice in trans healthcare to date have only focused on transition or HIV, rather than mental health or primary care.[9] In addition, primary care interventions for trans and non-binary patients have focused on specific areas including ensuring equitable screening access,[10] addressing gaps in secondary care services and on communication. There is much less evidence about healthcare access and patient experience for non-trans-specific services, and little on the epidemiology of long-term health conditions for trans and non-binary adults, despite these being central dimensions of the work of primary care, and making up the majority of all healthcare contacts.

Data collection within the NHS on trans and non-binary adults and their experiences of healthcare is lacking. To address this evidential need, NHS England is beginning to include questions asking about both gender and trans status in its surveys to support quality improvement programmes.[11][12] We used these data from the GP Patient Survey to answer the research question 'what are the demographic characteristics, health conditions, and healthcare experiences of trans and non-binary adults in England?'.

## METHODS

The methods for this cross-sectional secondary data analysis of the GP Patient Survey, including details of the survey data collection,[11] survey tools,[13] measures of gender and trans status, outcomes and outcome coding, statistical analysis, and patient and public involvement have been described in full in the preregistered study protocol[14] and are reproduced directly from this protocol in an edited form in this section. The protocol specifies three research questions; of these, RQ1 ('what are the demographic characteristics, health conditions and healthcare experiences of trans and non-binary adults in England?') is addressed in this work.[14]

### Data

In January 2021, the GP Patient Survey was sent by post to 2 408 303 adult patients registered with a GP in England, from 6694 GP practices, followed by an SMS reminder and two further postal mailings to initial non-responders. Patients from GP practices with historically low response rates are oversampled. Paper and online responses were possible.

Data were shared with the University of Cambridge under a data sharing agreement with NHS England and were not linked with any other GP practice or individual data.

## Survey measures

### Gender

In 2021, for the first time, the GP Patient Survey included revised questions covering sex, gender and gender reassignment in order for NHS England to meet its duties under the Equality Act[15] to collect data and address health inequalities in relation to both sex and gender reassignment.

The new questions were developed in consultation with NHS England and stakeholders and were tested in interviews with patients, including trans and non-binary patients, during September 2020. In addition, the questionnaire was reviewed by the Plain English Campaign to meet Plain English criteria; a set of principles designed to ensure information is presented clearly.

The two included questions ask:

'Which of the following best describes you?' with response options 'Female', 'Male', 'Non-binary', 'Prefer to self-describe' and 'Prefer not to say'; and 'Is your gender identity the same as the sex you were registered at birth?' with response options 'Yes', 'No' and 'Prefer not to say'.

### Long-term physical and mental health conditions

Respondents were asked 'Which, if any, of the following long-term conditions do you have?' with 17 response options 'Alzheimer's disease or other cause of dementia', 'Arthritis or ongoing problem with back or joints', 'Autism or autism spectrum condition', 'Blindness or partial sight', 'A breathing condition such as asthma or COPD', 'Cancer (diagnosis or treatment in the last 5 years)', 'Deafness or hearing loss', 'Diabetes', 'A heart condition such as angina or atrial fibrillation', 'High blood pressure', 'Kidney or liver disease', 'A learning disability', 'A mental health condition', 'A neurological condition such as epilepsy', 'A stroke (which affects your day to day life)', 'Another long-term condition or disability' and 'I do not have any long term conditions'.

### Patient experience

In the GP Patient Survey, questions ask about experiences of primary care across five domains of healthcare quality: experience before making an appointment, access, continuity, communication and overall experiences of care, with Likert scale response options. For analysis, these were categorised into binary (positive/negative) indicators for reporting, in line with national reporting.[5] Full question wording for the 18 survey items included in this analysis and coding of response options are presented in online appendix table 1.

### Sociodemographic characteristics

We included age, ethnicity, sexual orientation coded from survey responses and a small area (postcode) based measure of socioeconomic deprivation (the index of multiple deprivation) categorised into five groups based on quintile defining cut points.

## Statistical analysis

In preliminary analyses as per protocol,[14] we explored variation by region (the nine government office regions from England) and found little variation across the country except for a slightly higher number of trans and non-binary adults in London, and no impact of region in adjusted models, and so for simplicity we did not include full regional details in our final reporting. We also explored variation across all analyses considering trans and non-binary respondents separately. There are differences in the sizes but not directions of differences between these groups, which warrant further investigation. In this report, in an approach acceptable to the patient and public involvement panel with whom this work was developed,[14] for clarity, we consider all trans and non-binary responses, and additionally responses from people who prefer to self-describe their gender, together in a single group.

To explore demographics of the trans and non-binary population, we described the age, gender, ethnicity, deprivation and sexual orientation of trans and non-binary and all other survey respondents using weighted percentages.

Responses are weighted to the age and gender profile of each GP practice from which responses are sampled, based on information from the sampling frame, to account for design, non-response and calibration to the population of eligible patients. Full details are in the study technical guidance.[11] As GP registration in England is almost universal, these weighted estimates can be considered to be nationally representative of the population of England.

To explore long-term health conditions, we described the unadjusted weighted percentage of people living with each of 15 physical or mental health conditions, and in adjusted analysis, estimated the OR for reporting each condition for trans and non-binary adults, compared with all other survey respondents, after adjustment for age, deprivation and ethnicity. Because adjusted analyses include age and gender in the models, these are not weighted. In preliminary analyses, we additionally considered adjustment for GP practice using a random effect (in order to estimate disparities within GP practices, rather than fixed effect estimates for the population); this had no impact on these models in terms of either effect size, or statistical significance, and so we used fixed effect for population-level estimates, accounting for the GP practice based sampling using cluster robust SEs.

We also explored whether there was evidence for variation in the relationship between being trans and non-binary and each of the 15 long-term health conditions varied by age, ethnicity, deprivation, sexual orientation and region using an integration term in each model. There was evidence of heterogeneity in the association with trans status for 10 out of the 15 conditions in different age groups and so we estimated adjusted prevalence stratified by age using recycled predictions from these models, adjusting to the 'average' deprivation and ethnicity of all survey respondents.

To explore the healthcare experience of the trans and non-binary population, we calculated the unadjusted weighted percentage of people reporting a positive experience for the 18 patient items. In preliminary analyses, we again explored both national (fixed) and within practice (random effect) models. Estimates and standard errors were again the same from both approaches, and estimates from fixed effect models are presented here.

All analyses were carried out using Stata V.15.3 Statistical Software (2017) StataCorp.

### Patient and public involvement

The protocol for this work was developed in collaboration with a trans, non-binary and queer patient and public involvement panel. We met an additional two times online and once in person during this project.

## RESULTS

In total, 850 206 survey responses were received to the 2021 GP Patient Survey (35% response rate, with 36% of responses online). These included 9515 where neither gender nor trans status were reported (ie, either 'prefer not to say' or missing responses to both questions), who were excluded from all analyses. Of the 840 691 included responses, there were 6333 trans and non-binary respondents in total (0.9%) who selected either non-binary, or a self-definition for their gender, and/or those who affirmed their gender identity was different from their sex registered at birth and were included in descriptive analyses. We included only responses with complete age, ethnicity and deprivation in multivariable analyses; there were 827 696 participants, of whom 6091 were trans and/or non-binary. Respondents with missing outcome data were excluded from the analyses for long-term health conditions and patient experience on an outcome by outcome basis.

### Demographic characteristics

Trans and non-binary adults were younger, more likely to be from Asian, black, mixed or other ethnic groups, less likely to be heterosexual and are more likely to live in more deprived parts of the country (table 1, online appendix table 2, and for the multivariable analysis sample only in online appendix table 3).

### Long-term conditions

The weighted, unadjusted, percentages of each long-term condition among trans and non-binary respondents and all other survey respondents are presented in table 2, with 39.3% of trans and non-binary respondents and 38.3% of all other respondents reporting no long-term health conditions. However, after adjustment for age, ethnicity and deprivation, overall, for 10 out of the 15 long-term conditions, trans and non-binary adults reported higher prevalence, with inequalities largest for autism: OR (95% CI), 5.8 (5.0 to 6.6), dementia: 3.1 (2.5 to 3.9), learning disabilities: 2.8 (2.4 to 3.2) and mental health:

Table 1  Respondent characteristics (all respondents, n=840 691)

| | All trans and non-binary respondents (n, %) | All other survey respondents (n, %) |
|---|---|---|
| **All respondents (n=840 691)** | 6333 | 834 358 |
| **Gender (n=835 561)** | | |
| Female | 1708 (28.2) | 468 958 (56.5) |
| Male | 1971 (32.6) | 359 266 (43.3) |
| Non-binary | 1220 (20.2) | |
| Prefer to self-describe | 1047 (17.3) | |
| Prefer not to say | 103 (1.7) | 1288 (0.2) |
| **Trans status (n=834 746)** | | |
| Gender identity the same as sex registered at birth | 957 (15.3) | 825 209 (99.6) |
| Gender identity different from sex registered at birth | 4642 (74.4) | |
| Prefer not to say | 644 (10.3) | 3294 (0.4) |
| **Age (years) (n=833 526)** | | |
| 16–24 | 628 (10.1) | 41 162 (5.0) |
| 25–34 | 838 (13.5) | 69 764 (8.4) |
| 35–44 | 1109 (17.9) | 100 108 (12.1) |
| 45–54 | 1155 (18.6) | 137 231 (16.6) |
| 55–64 | 1080 (17.4) | 174 512 (21.1) |
| 65–74 | 827 (13.3) | 172 155 (20.8) |
| 75–84 | 573 (9.2) | 132 384 (16.0) |
| **Ethnicity (n=834 261)** | | |
| White | 3343 (54.0) | 702 888 (84.9) |
| Asian | 176 (2.8) | 10 780 (1.3) |
| Black | 1467 (23.7) | 70 305 (8.5) |
| Mixed | 482 (7.8) | 27 612 (3.3) |
| Other | 727 (11.7) | 16 481 (2.0) |
| **Sexual orientation (n=820 113)** | | |
| Heterosexual | 2682 (46.1) | 754 746 (92.7) |
| Lesbian/gay | 310 (5.3) | 11 452 (1.4) |
| Bisexual | 462 (7.9) | 8186 (1.0) |
| Other | 752 (12.9) | 6235 (0.8) |
| Prefer not to say | 1614 (27.7) | 33 674 (4.1) |
| **Deprivation (n=840 691)** | | |
| Most deprived | 2211 (34.9) | 162 804 (19.5) |
| 2 | 1650 (26.1) | 167 119 (20.0) |
| 3 | 1117 (17.6) | 172 925 (20.7) |
| 4 | 791 (12.5) | 169 938 (20.4) |
| Least deprived | 561 (8.9) | 161 387 (19.3) |

2.0 (1.9 to 2.2) (table 2). Trans and non-binary respondents were 30% less likely to report no long-term conditions after adjustment for age, deprivation and ethnicity, 0.7 (0.7 to 0.8).

Adjusted percentages stratified by age are presented in figure 1 and online appendix table 4 for model estimates, including confidence intervals; in general, the differences between trans and non-binary adults (black lines) and all other survey respondents (grey lines) are smaller than the differences between conditions. Trans and non-binary adults are presenting to primary care with similar health profiles to those of all other survey respondents; the age-related patterns of long-term conditions and variation between conditions is similar across both trans and non-binary respondents and all other survey respondents.

However, despite these overarching similarities in prevalence, on inspection, there are four different groups of age-related patterns across conditions. First, both autism and autistic spectrum conditions, and mental health problems, have higher prevalence at younger ages, and prevalence among trans and non-binary respondents are particularly high at younger ages with absolute differences reducing with age. Second, for dementia, learning disability, blindness and neurological conditions including epilepsy, adjusted prevalence in trans and non-binary adults follows the same pattern as all other survey respondents, but prevalence shifted upwards with higher prevalence at each age, although absolute differences are small. Third, for stroke, diabetes and kidney or liver problems, the disparities increase with age, with trans and non-binary adults having increasingly higher prevalence compared with all other survey respondents at older age. Finally, for arthritis, hypertension, cancer, breathing problems, heart conditions and deafness, adjusted prevalence in trans and non-binary adults at older ages is lower than prevalence for all other survey respondents (figure 1, online appendix table 4 for model estimates and online appendix figures 1-3 for versions of figure 1 with rescaled axes for clarity).

## Patient experience

After adjusting for age, ethnicity, deprivation and practice, we found that trans and non-binary adults reported slightly poorer overall primary care experiences than all other survey respondents (table 3) 0.8 (0.8 to 0.9) and across access and overall experience items reported higher numbers of both very good and very poor experiences (table 4); however, there were stronger differences across different domains of patient experience.

There was no evidence of a difference in overall experience of making an appointment, 1.0 (1.0 to 1.1), with only small differences in experiences across different domains of access (OR range 0.8–1.2 across items); however, for continuity, trans and non-binary adults were much more likely to report that they had a preference for a particular GP 1.7 (1.6 to 1.8), with no difference compared with all other survey respondents in whether they were able to see this preferred GP or not 1.1 (1.0 to

Table 2 Long-term health conditions (735 078 responses, 5110 trans and non-binary)

| | All trans and non-binary respondents | All other survey respondents | Adjusted for age, deprivation and ethnicity | |
| --- | --- | --- | --- | --- |
| | (n, weighted %) | (n, weighted %) | OR (95% CI) | P value |
| Autism or autism spectrum condition | 227 (4.4) | 4236 (0.6) | 5.8 (5.0 to 6.6) | <0.0001 |
| Alzheimer's disease or other cause of dementia | 96 (1.9) | 5471 (0.7) | 3.1 (2.5 to 3.9) | <0.0001 |
| A learning disability | 204 (4.0) | 7607 (1.0) | 2.8 (2.4 to 3.2) | <0.0001 |
| A mental health condition | 891 (17.4) | 66 363 (9.1) | 2.0 (1.9 to 2.2) | <0.0001 |
| A stroke (which affects your day to day life) | 81 (1.6) | 7638 (1.0) | 1.8 (1.4 to 2.2) | <0.0001 |
| Blindness or partial sight | 118 (2.3) | 11 955 (1.6) | 1.7 (1.4 to 2.0) | <0.0001 |
| Kidney or liver disease | 156 (3.1) | 17 038 (2.3) | 1.4 (1.2 to 1.6) | <0.0001 |
| A neurological condition such as epilepsy | 116 (2.3) | 13 855 (1.9) | 1.3 (1.1 to 1.6) | 0.003 |
| Deafness or hearing loss | 287 (5.6) | 55 229 (7.6) | 1.2 (1.1 to 1.4) | 0.002 |
| Diabetes | 657 (12.9) | 72 176 (9.9) | 1.2 (1.1 to 1.4) | <0.0001 |
| Another long-term condition or disability | 645 (12.6) | 94 872 (13.0) | 1.1 (1.0 to 1.2) | 0.009 |
| A heart condition such as angina or atrial fibrillation | 279 (5.5) | 55 841 (7.6) | 1.0 (0.9 to 1.2) | 0.74 |
| A breathing condition such as asthma or COPD | 523 (10.2) | 89 611 (12.3) | 1.0 (0.9 to 1.0) | 0.28 |
| Arthritis or ongoing problem with back or joints | 865 (16.9) | 166 498 (22.8) | 0.9 (0.9 to 1.0) | 0.18 |
| High blood pressure | 867 (17.0) | 158 434 (21.7) | 0.9 (0.9 to 1.0) | 0.055 |
| Cancer (diagnosis or treatment in the last 5 years) | 121 (2.4) | 31 555 (4.3) | 0.9 (0.7 to 1.0) | 0.092 |
| I do not have any long-term conditions | 2006 (39.3) | 279 314 (38.3) | 0.7 (0.7 to 0.8) | <0.0001 |

1.2). Disparities in experiences of interpersonal communication were greatest. After adjustment, trans and non-binary respondents were about half as likely to report a positive experience than all other survey respondents. Trans and non-binary respondents were more likely to report having mental health needs during their appointment 2.1 (1.9 to 2.2); however, among people with mental health needs, trans and non-binary people were less likely to report that these needs had been met (table 3).

## DISCUSSION

Trans and non-binary adults in England responding to the GP Patient Survey reported poorer health and poorer experiences in primary care than all other survey respondents. This research adds up-to-date evidence in an area where our understanding of the population demographics and healthcare need is still emerging, from a large, nationally representative sample. This analysis highlights that autism and autistic spectrum conditions, dementia, learning disability and mental health problems are the conditions where disparities are greatest, with variation by age. In healthcare experience, disparities are much greater in interpersonal communication than access. Additionally, trans and non-binary adults report much higher preference for continuity of care than all other survey respondents, with no evidence of any differences in being able to see or speak to a preferred GP. For measures of overall primary care experience, and access,

both more very positive and more very negative experiences were reported.

These results, considering both health outcomes and patient experience in primary care, are particularly important in the context of evidence that healthcare outcomes among trans and gender diverse individuals are better when accessing a primary care provider who is knowledgeable about trans health.[16] They also provide insight for understanding the demographics of the trans population in primary care and the health needs of this group.

Our finding that trans and non-binary adults are over-represented among adults in autism and autistic spectrum conditions, dementia, learning disability and mental health problems in primary care is consistent with previous research that found higher rates of autism, other neurodevelopmental and psychiatric diagnoses in trans and gender diverse study participants.[17] Our findings of poorer healthcare experiences from trans and non-binary adults also reflect findings from earlier work.[2] Our analyses build on and develop the evidence base from these previous studies, with population-based nationally representative estimates, and provide additional insight about the size of these disparities for most long-term health conditions.

This work provides novel insight into how disparities in long-term health conditions among trans and non-binary individuals vary with age. The four different age-related

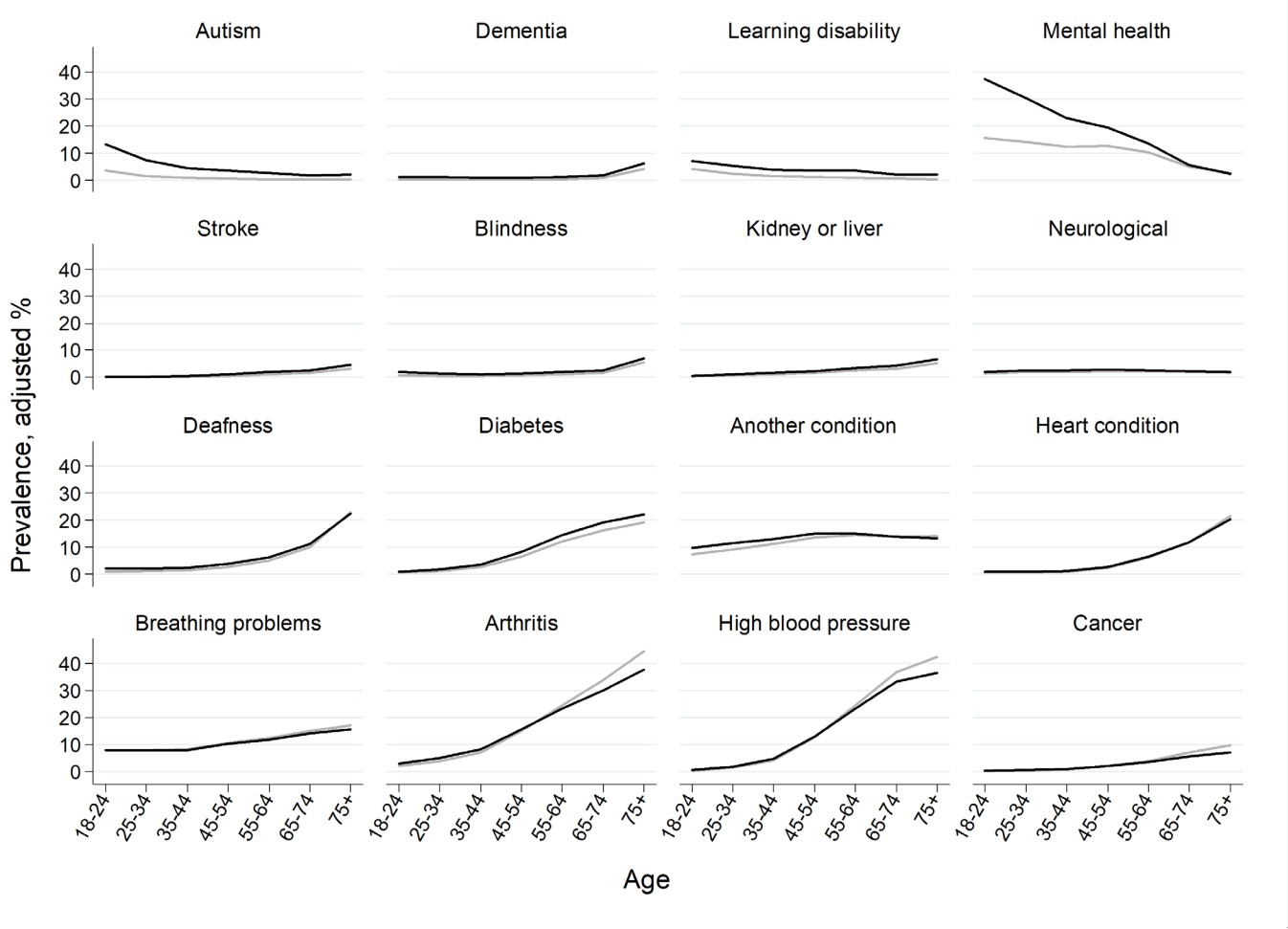

**Figure 1** Long-term condition prevalence, stratified by age. Black line: all trans and/or non-binary respondents; grey: all other survey respondents.

patterns for long-term conditions point to heterogeneous and potentially intersecting factors driving some of the differences seen. It is interesting to note that this variation in age-related patterns between long-term health conditions is not seen when considering inequalities in long-term health conditions by socioeconomic deprivation, for which inequalities consistently increase with age,[18] nor by sexual orientation, where disparities across almost all long-term health conditions are greatest at younger ages and decrease at older ages.[19] For trans and non-binary adults, the drivers of differences are likely to be more condition specific. These patterns are likely, among other possible mechanisms, to be the result of complex inter-actions between minority stress, behavioural risk factors and biological effects of exogenous hormones, as well as sexual orientation, socioeconomic status and healthcare access. In addition, trans and non-binary people are not a single homogenous group, and the patterns observed in long-term health conditions may be in part driven by differences across subgroups.

From an epidemiological perspective, it is likely that age, period and cohort effects may all be important in understanding these relationships. For example, the impact of legislation including the Local Government Act (Section 28) in 1988,[20] Gender Recognition Act in 2004[21] and the Equalities Act in 2010[15] are likely to have had an impact for trans adults during these years (period effects). Mental health may have been affected by external factors such as transphobic media focus in more recent years. Long-term gender-affirming therapies may have specific health impacts over time (age effects), and we speculate but cannot confirm that this may be part of the explanation for differences in prevalence of some long-term conditions that are hormonally mediated. Higher mortality among trans and non-binary adults may potentially also have an impact on long-term condition prevalence at older ages, but this cannot be evaluated in this cross-sectional analysis.

We note that trans and non-binary survey respondents were much younger, on average than all other survey respondents, which may suggest that people are more comfortable coming out as trans and at younger ages than previously.[22] The experiences of people who are trans and in their late teens/early 20s in 2021 may be very different from a similarly aged cohort in the past. The much higher prevalence of autism and autistic spectrum

**Table 3** Patient experience, coding for public reporting

| Experience items | Number of responses | All trans and non-binary respondents (n, weighted % positive response) | All other survey respondents (n, weighted % positive response) | OR (95% CI) adjusted for age, deprivation, ethnicity | P value |
|---|---|---|---|---|---|
| **Overall experience** | | | | | |
| Overall (positive) experience | 816 702 | 80.3 | 85.7 | 0.8 (0.8 to 0.9) | <0.0001 |
| Overall (positive) experience of making an appointment | 752 868 | 71.4 | 73.9 | 1.0 (1.0 to 1.1) | 0.84 |
| **Before trying to make an appointment** | | | | | |
| *Tried self-management* | 742 768 | 30.4 | 31.7 | 0.9 (0.8 to 0.9) | <0.0001 |
| *Asked friends or family* | 742 768 | 8.4 | 7.7 | 1.0 (0.9 to 1.1) | 0.53 |
| *Tried online, telephone or other NHS services* | 742 768 | 25.0 | 16.8 | 1.3 (1.3 to 1.4) | <0.0001 |
| *Tried online or other non-NHS services* | 742 768 | 51.8 | 58.0 | 0.9 (0.8 to 0.9) | <0.0001 |
| **Access** | | | | | |
| Found GP practice website easy to use | 389 957 | 70.9 | 77.1 | 0.8 (0.8 to 0.9) | <0.0001 |
| *Tried to make an appointment in the last 6 months* | 777 102 | 66.5 | 64.8 | 1.0 (1.0 to 1.1) | 0.69 |
| Getting through on the phone | 789 116 | 69.7 | 72.5 | 1.0 (0.9 to 1.1) | 0.82 |
| Helpful receptionists | 795 619 | 86.9 | 90.9 | 0.9 (0.8 to 0.9) | 0.0001 |
| Offered choices when booking appointment | 570 565 | 74.9 | 71.4 | 1.2 (1.2 to 1.3) | <0.0001 |
| Satisfied with appointment times available | 715 175 | 73.0 | 71.0 | 1.2 (1.1 to 1.3) | <0.0001 |
| Offered an acceptable appointment | 749 975 | 87.4 | 90.7 | 0.8 (0.7 to 0.9) | <0.0001 |
| Satisfied with appointment offered | 695 269 | 79.8 | 83.5 | 1.0 (0.9 to 1.0) | 0.18 |
| Remote appointment (telephone or online) | 648 288 | 53.9 | 49.2 | 1.1 (1.1 to 1.2) | <0.0001 |
| **Continuity** | | | | | |
| *Have a preferred GP* | 776 310 | 64.2 | 50.2 | 1.7 (1.6 to 1.8) | <0.0001 |
| Able to see preferred GP | 216 333 | 59.5 | 58.8 | 1.1 (1.0 to 1.2) | 0.12 |
| **Communication** | | | | | |
| Involved in decisions about care and treatment | 668 530 | 85.0 | 93.7 | 0.5 (0.5 to 0.6) | <0.0001 |
| *Had mental health needs in last appointment* | 604 671 | 76.2 | 55.5 | 2.1 (1.9 to 2.2) | <0.0001 |
| Mental health needs recognised and understood | 336 574 | 77.6 | 87.3 | 0.7 (0.6 to 0.7) | <0.0001 |
| Confidence and trust | 744 575 | 88.7 | 96.3 | 0.4 (0.4 to 0.4) | <0.0001 |
| Needs were met | 745 767 | 88.1 | 95.2 | 0.5 (0.5 to 0.5) | <0.0001 |

Experience items marked in *italics* refer to preferences or choices, rather than evaluation of care quality. These questions as well as evaluative items are presented side by side here for clarity across the patient journey.
NHS, National Health Service.

conditions and mental health problems among young trans adults compared with older trans adults may reflect these cohort effects as well. Longitudinal population-based data collections are needed to understand causal relationships and properly disentangle the impacts of

coming out at different ages on our understanding of these health impacts.[23]

Although the patterns of age-specific long-term health condition prevalence identified provide some insight into disparities experienced by trans and non-binary adults, it

**Table 4** Patient experience – very positive and very negative responses

| | Adjusted % (95%CI) endorsing the most positive response option | | Adjusted % (95%CI) endorsing the most negative response option | |
|---|---|---|---|---|
| | Trans and non-binary respondents | All other survey respondents | Trans and non-binary respondents | All other survey respondents |
| **Overall experience** | | | | |
| Overall experience | 54.4 (53.1–55.6) | 53.3 (53.2–53.5) | 3.3 (2.9–3.7) | 1.9 (1.8–1.9) |
| Overall experience of making an appointment | 41.2 (39.8–42.5) | 36.3 (36.2–36.4) | 5.8 (5.2–6.4) | 4.6 (4.5–4.6) |
| **Access** | | | | |
| Found GP practice website easy to use | 31.9 (30.3–33.5) | 26.7 (26.6–26.8) | 9.1 (8.2–10.1) | 6.8 (6.7–6.9) |
| Getting through on the phone | 33.7 (32.5–35.0) | 25.3 (25.2–25.4) | 9.8 (9.1–10.5) | 9.5 (9.5–9.6) |
| Helpful receptionists | 56.0 (54.7–57.2) | 50.3 (50.2–50.4) | 3.2 (2.8–3.6) | 2.4 (2.3–2.4) |
| **Communication** | | | | |
| Involved in decisions about care and treatment | 60.4 (59.0–61.8) | 64.0 (63.9–64.1) | 11.5 (10.7–12.4) | 6.3 (6.2–6.3) |
| Confidence and trust | 68.3 (67.1–69.6) | 74.1 (74.0–74.2) | 8.7 (8.0–9.5) | 3.7 (3.7–3.8) |
| Needs were met | 63.7 (62.3–65.0) | 67.7 (67.6–67.8) | 9.1 (8.4–9.8) | 4.9 (4.8–4.9) |

is also important to highlight that for most conditions, differences in patterns of prevalence between trans and non-binary and all other survey respondents are smaller than differences in prevalence between conditions. This finding highlights that for long-term condition management in primary care the epidemiology of conditions among trans and non-binary adults within a practice is likely to be similar to all other patients within the practice. Standard best practice for the management of long-term conditions, unrelated to trans status, will likely form a large part of the work of GPs with trans and non-binary patients in their practice. Previous work has highlighted that guidelines beyond transition or HIV care still need development.[9] Research areas that underpin long-term condition management, such as ensuring risk prediction models for long-term conditions can be correctly implemented for trans and non-binary patients, will remain a priority.[24]

The areas where there are substantial differences in prevalence, however, are for autism and autistic spectrum conditions and mental health problems, which are both very high among young trans and non-binary adults. In 2016, the Royal College of General Practitioners produced a position statement and best practice guidance, making commitments to enabling access for patients on the autistic spectrum to primary care and recognising the additional needs of those with autism.[25] Similar guidance for mental health problems was published in 2017.[26] These guidelines are approaches which, again although not trans-specific, provide practical strategies that may be worth considering for addressing disparities in the poorer healthcare experiences of trans and non-binary patients in primary care.

Our findings describing the patient experience of trans and non-binary patients in primary care provide further insight into where quality improvement work could be directed. The relatively small disparities in access compared with the much larger inequalities in communication experienced by trans and non-binary adults are one such example. Resources are beginning to be developed to support best practice for primary care for communication with trans and non-binary adults in primary care. The 'healthtalk' resource includes interviews with 50 trans and gender diverse young people highlighting individual reflections on how experiences of healthcare could be improved.[27] Collections of individual experiences provide important insight where evidence-based best practice guidelines are still developing.

The much higher number of trans and non-binary patients who have a preference for continuity is also worth highlighting. Some GPs are regarded by the trans community as providing good trans healthcare and therefore more trans people may register with these GPs[28] while surgeries where care is poorer may lose their trans patients. Some trans and non-binary people report having excellent GPs with high levels of training and understanding of trans patients while many do not; this may explain our finding of both very positive and very negative experiences. Our findings that trans and non-binary adults in England are more likely to be living with long-term health conditions again highlight the importance of continuity.[29] It also highlights that all GPs, regardless of location or special interest, are likely to encounter trans and non-binary patients and will be supporting particular needs relating to long-term health conditions. Where evidence for this work is lacking, further dedicated studies are warranted, with sufficient funding and coproduction with the trans and non-binary community.

It is important to identify that there are limitations to the findings from this research; several of these have

been discussed in the protocol.[14] From a survey methods perspective, non-differential misclassification of trans and non-binary adults is one concern that has been recognised recently in the development of tools to collect gender and trans status in the NATSAL 4 study.[30 31] Although the number of people incorrectly reporting a trans or non-binary identity will be small, they may form a non-trivial number of trans and non-binary respondents overall. This is one possible explanation for the association seen with dementia, where numbers are low, and inaccuracies (associated with cognitive decline) in inaccurately endorsing a trans and non-binary response may be higher. However, the consistency of the findings from this work with previous studies where there were interviewer-led surveys (including, eg, the finding of higher endorsement of both very good and very poor patient experience items)[32] give some reassurance that although this is a concern the impact may be low. A second limitation is the exclusion of respondents who did not respond or responded 'prefer not to say' to questions about gender and trans-status; it is a concern that these responses may differentially include people who are trans and non-binary. Previous analyses of the GP Patient Survey focusing on sexual orientation found poorer experiences in these groups.[33] It is also worth highlighting that these findings are limited to trans and non-binary patients who are registered with a GP, and the population who are not registered, or who access primary care only through private GP services such as Babylon,[28] will not be included.

Despite these limitations, the collection of trans status and an inclusive gender question in the GP Patient Survey is an important step towards the generation of evidence for the development of best practice guidance for trans and non-binary patients. Following guidance on delivering good care and communication for those with autism and autistic spectrum conditions and mental health problems, both of which are prevalent among young trans and non-binary adults, is one way some of the inequalities identified may be addressed. Long-term condition epidemiology, although with some variation between conditions, is broadly similar in both trans and non-binary adults and all other survey respondents. Good primary care, not just good trans healthcare, is important to address the disparities identified.

**Author affiliations**
[1]Primary Care Unit, Department of Public Health and Primary Care, University of Cambridge, Cambridge, UK
[2]Barts Cancer Institute, Queen Mary University of London, London, UK
[3]Tavistock and Portman NHS Foundation Trust, London, UK
[4]Department of Public Health & Primary Care, University of Cambridge, Cambridge, UK
[5]NHS England, London, UK
[6]University of Cambridge, Cambridge, UK
[7]Lifestrong, Wolverhampton, UK

**Acknowledgements** We would like to thank Roary Neat and all other members of the public involvement panel who contributed to this work.

**Contributors** CLS carried out the analysis, wrote the first draft of the paper and is guarantor. CLS, JL, AMM, MR, JS and RD developed the analysis protocol. All authors read, critically revised and reviewed the final draft.

**Funding** CLS holds a Career Development Fellowship (award number: MH041) funded as part of the Three NIHR Research Schools Mental Health Programme. AMM is funded by the EU/EFPIA Innovative Medicines Initiative Joint Undertaking BigData@Heart grant 116074. This research was supported by core funding from the: British Heart Foundation (RG/13/13/30194; RG/18/13/33946) and NIHR Cambridge Biomedical Research Centre (BRC-1215-20014). JL's work for this study is supported by the Wellcome Trust as part of the Wellcome Trust PhD programme for Primary Care Clinicans (grant number 203921/Z/16/Z).

**Disclaimer** The views expressed are those of the author(s) and not necessarily those of the NIHR or the Department of Health and Social Care.

**Competing interests** None declared.

**Patient and public involvement** Patients and/or the public were involved in the design, or conduct, or reporting, or dissemination plans of this research. Refer to the Methods section for further details.

**Patient consent for publication** Not applicable.

**Ethics approval** Not applicable.

**Provenance and peer review** Not commissioned; externally peer reviewed.

**Data availability statement** Data may be obtained from a third party and are not publicly available.

**ORCID iDs**
Catherine L Saunders http://orcid.org/0000-0002-3127-3218
Alison Berner http://orcid.org/0000-0002-1132-0275
Robbie Duschinsky http://orcid.org/0000-0003-2023-5328

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
