## [Reviewer comments · BMJ Open]

ARTICLE DETAILS

TITLE (PROVISIONAL)	Demographic characteristics, long-term health conditions, and healthcare experiences of 6,333 trans and non-binary adults in England: nationally representative evidence from the 2021 GP Patient Survey
AUTHORS	Saunders, Catherine; Berner, Alison; Lund, Jenny; Mason, Amy M; Oakes-Monger, Tash; Roberts, Meg; Smith, Jack; Duschinsky, Robbie

VERSION 1 – REVIEW

REVIEWER	McManus, Sally National Centre for Social Research, London, SRU
REVIEW RETURNED	29-Oct-2022

GENERAL COMMENTS	This paper profiles the demographics, health, and primary care experiences of a large, recent, national sample of adults who identify as trans and/or non-binary, and compares this profile with that of the rest of the adult population. This makes novel and very appropriate use of the NHS' GP Patient Survey, a well-run, probability sample national survey of people registered as patients in primary care in England. Results are presented for well over 6,000 trans and non-binary adults across the entire adult age range, with weights applied to address non-response. I struggle to think on any data source on this population of comparable size and robustness. PPI engagement occurred at multiple stages of the project and informed analysis decisions. The paper is extremely clearly written, with clear use of charts. Results have great relevance, both for primary care and for more widely understanding the size and needs of this population. In terms of health, the high rates of autism identified is notable, as is dementia and learning disability. In terms of primary care, the greater preference for continuity of provider and the greater concerns around interpersonal communications really stand out. The analyses looking at interactions with age are important and clearly presented. Interactions with socioeconomic factors - e.g. IMD - would be of interest but beyond the scope of this paper which is both already dense and also specified in a preregistered protocol. The patterns of associations with ethnicity are also noteworthy. In summary - really clear and important use of an impressive dataset, with clear implications for general policy and specifically for primary care practice.
---

REVIEWER	Teece, Lucy University of Leicester, Department of Health Sciences
REVIEW RETURNED	04-Nov-2022

GENERAL COMMENTS

This is an important study in which the experiences of trans and non-binary adults throughout primary care in England are summarised, redressing the deficit of data in this area. The research question is well-defined and evaluated successfully, and the team make important steps towards informing of trans and non-binary experiences within primary care in England. The research was advised by a trans, non-binary and queer patient and public involvement panel.

I have listed below some suggested edits to the article, which I believe will improve the readers understanding and ability to reproduce the study:

Major items:

1. Given the readership of BMJ Open and that social contexts and terminology referring to trans and non-binary individuals varies greatly, definitions of 'trans' and 'non-binary' in this specific context could be important for many readers and would be useful if added to the introduction section.

2. There is currently insufficient detail in the provided manuscript to understand or evaluate the research methods used. Authors direct readers to a protocol that is not specific to this study, and addresses multiple research questions. The addition of important details to the main manuscript would reduce the burden on the reader, allowing this to be read as a stand-alone document. Brief details on the items suggested in the STROBE/RECORD reporting guidelines would be both beneficial and expected by the reader (study design, setting, participants, variables, data sources, bias, study size, quantitative variables, statistical methods, data access and cleaning methods, linkage). Please include details of methods for weighting and analysis.

3. There are several findings reported in the methods section that I believe would be more appropriate in the results section; included/excluded numbers, data-driven decisions (random effects and heterogeneity), and statements about the final data.

4. Please consider the addition of a flow diagram in the results section to show the impact of the inclusion and exclusion criteria on the derived study sample size. I believe this would allow easier visualisation by the reader.

5. Table 1, which reports characteristics of all respondents (not just those included in the study analysis), is important to answer the first objective of exploring the demographics of the trans and non-binary community. This could be improved with the addition of levels of missing information for key variables (age, ethnicity, deprivation) and by the reporting of column-wise percentages for these variables. While I acknowledge that row percentages may be more useful for the first 3 sections (All Respondents, Gender, Trans Status), column percentages are more useful and intuitive for the reader for other demographic variable rows and differences between small percentages (caused by low prevalence of trans/NB adults) are more difficult to interpret. I appreciate row-wise percentages may be preferred in some clinical contexts, so the authors may choose to include these in the appendix.

6. An additional supplementary table, limited to summarising the characteristics of those included in the analysis (827,696 total with

6,091 trans/NB) would be beneficial. In this case, demographic distributions would reflect the sample included in the analyses.

7. The author's decision whether to report a random or fixed effects model appears to have been decided by whether the approaches affected study estimates, "Estimates were the same from both approaches... fixed effects are reported here", rather than on whether the underlying mean response is expected to vary between GPs. Random effects often increase SEs (and hence CIs/P-values) while point estimates are expected to remain unchanged. Please include more information on their decision for reporting the fixed effects model.

8. On page 15: 'trans and non-binary respondents reported experiences of communication about 15 percentage points poorer'. It is not clear which results this statement relates to, as no differences are as large as 15 pp. Might this be an error (if so please correct) or have been derived from unreported analyses (if so please add details)?

Minor items:

Methods:

9. Please reference the STATA software where mentioned: <https://www.stata.com/support/faqs/resources/citing-software-documentation-faqs/>

Results:

10. Page 12 paragraph 2; "MH" is not defined/expanded.

Discussion:

11. In the 4th discussion paragraph, when possible reasons for differences in age-related patterns are given – "These patterns are likely the result of complex interactions between minority stress, behavioural risk factors, and biological effects of exogenous hormones, as well as sexual orientation and socioeconomic status." – perhaps more open wording might better convey the idea that the exact causes are not fully known by adding 'among others' to the end. Other likely causes could be due to increased GP engagement meaning increased chance of diagnosis.

12. Please expand on the potential impact of misclassification (given as a possible explanation for Alzheimer's results) on study results/findings.

Tables and figures:

13. Figure 1:

a. Please include a legend for the line assignment.

b. The continuous plot of a discrete scale could be misleading, especially with the jump to the 75+ cat (that may not be an even step-size). It may be less misleading to have a continuous x-axis with point estimates placed at the median age for that group. Alternatively, presenting disconnected points with error bars would be suitable.

14. Table 3: Suggest renaming "overall experience" to 'Positive Overall Experience' to make clear that the variable has been dichotomised (otherwise it is not clearly binary).

	15. Table 4: "Adjusted % endorsing the most positive response option / Adjusted % endorsing the most negative response option" – should these be 'weighted percentage'? The review was completed by Mr Angus Jennings and Dr Lucy Teece.
--	--

REVIEWER	Gianelis, Kristin Frontier Nursing University
REVIEW RETURNED	15-Nov-2022

GENERAL COMMENTS	Thank you for your attention to this important topic. In the introductory paragraph (page 5 of 35), focus is placed on the primary care provider's responsibility towards addressing long-term health conditions. The care of trans and non-binary persons, which is the true focus of this article, is only introduced in the final sentence of this paragraph. Consider revising this first paragraph to better present the topic and place emphasis on the population of focus rather than the setting. Additionally, noting care of trans and non-binary individuals after discussion of "long-term health conditions" may lead a reader to infer that gender identity is a "health condition" by itself. Page 5, line 19 - "gender reassignment" is not frequently used in US terminology. Unclear if this commonplace if UK scientific literature. In the discussion section (pages 16-19), results were well presented and explored. Suggest further clarification that the study findings do not suggest a causative relationship between trans and non-binary gender identities and the various neuropsychological conditions where greatest disparities existed. While this may seem unnecessary given the clear statistics and discussion provided, in the current sociopolitical climate, tremendous care should be taken to protect vulnerable populations from harmful attempts at misrepresenting data.
--

VERSION 1 – AUTHOR RESPONSE

Reviewer: 1

This reviewer was entirely supportive of our paper which is very pleasing to read.

Reviewer: 2

Comments to the Author:

This is an important study in which the experiences of trans and non-binary adults throughout primary care in England are summarised, redressing the deficit of data in this area. The research question is well-defined and evaluated successfully, and the team make important steps towards informing of trans and non-binary experiences within primary care in England. The research was advised by a trans, non-binary and queer patient and public involvement panel.

Thank you for this positive assessment of our work

I have listed below some suggested edits to the article, which I believe will improve the readers understanding and ability to reproduce the study:

Major items:

1. Given the readership of BMJ Open and that social contexts and terminology referring to trans and non-binary individuals varies greatly, definitions of 'trans' and 'non-binary' in this specific context could be important for many readers and would be useful if added to the introduction section.

2. There is currently insufficient detail in the provided manuscript to understand or evaluate the research methods used. Authors direct readers to a protocol that is not specific to this study, and addresses multiple research questions. The addition of important details to the main manuscript would reduce the burden on the reader, allowing this to be read as a stand-alone document. Brief details on the items suggested in the STROBE/RECORD reporting guidelines would be both beneficial and expected by the reader (study design, setting, participants, variables, data sources, bias, study size, quantitative variables, statistical methods, data access and cleaning methods, linkage). Please include details of methods for weighting and analysis.

Thank you for this helpful review. We have substantially expanded the methods section and hope that this now is much clearer. The new appendix table 1 includes details of the quantitative variable coding and the statistical methods are now highlighted with a sub heading and expanded. More details of the weighting approach have been included.

3. There are several findings reported in the methods section that I believe would be more appropriate in the results section; included/excluded numbers, data-driven decisions (random effects and heterogeneity), and statements about the final data.

We have moved the details about included and excluded numbers to a new paragraph at the start of the results section, and have included more detail about the final data sets for both descriptive and multivariable analysis.

We have added more methodological detail into the analysis sections in the methods about both the random effects models and about heterogeneity

4. Please consider the addition of a flow diagram in the results section to show the impact of the inclusion and exclusion criteria on the derived study sample size. I believe this would allow easier visualisation by the reader.

We have moved the details of the flow of study participants to the start of the results section. We think that the flow of study participants in this new section at the start of the results now makes this derivation of the study sample clear. We would be happy to revisit this and add a further diagram if the reviewers still think this is necessary, but hope that this reworking is now clear as it is.

5. Table 1, which reports characteristics of all respondents (not just those included in the study analysis), is important to answer the first objective of exploring the demographics of the trans and non-binary community. This could be improved with the addition of levels of missing information for key variables (age, ethnicity, deprivation)

Sorry, yes, I think this was an error missed in the previous version of this table. The updated has details of the sample size for all the key variables now.

... and by the reporting of column-wise percentages for these variables. While I acknowledge that row percentages may be more useful for the first 3 sections (All Respondents, Gender, Trans Status), column percentages are more useful and intuitive for the reader for other demographic variable rows and differences between small percentages (caused by low prevalence of trans/NB adults) are more difficult to interpret. I appreciate row-wise percentages may be preferred in some clinical contexts, so the authors may choose to include these in the appendix.

Thanks for this suggestion. We have changed Table 1 to column percentages and have put the row percentages into Appendix Table 2.

6. An additional supplementary table, limited to summarising the characteristics of those included in the analysis (827,696 total with 6,091 trans/NB) would be beneficial. In this case, demographic distributions would reflect the sample included in the analyses.

Thanks. We have included this as new Appendix Table 3.

7. The author's decision whether to report a random or fixed effects model appears to have been decided by whether the approaches affected study estimates, "Estimates were the same from both approaches... fixed effects are reported here", rather than on whether the underlying mean response is expected to vary between GPs. Random effects often increase SEs (and hence CIs/P-values) while point estimates are expected to remain unchanged. Please include more information on their decision for reporting the fixed effects model.

We have included more information about this in the methods section, in both the long-term health condition and patient experience outcomes (fourth and sixth paragraphs of the statistical analysis section). We have clarified that the similarities in estimates from both modelling approaches were for both estimates and measures of precision. We used cluster robust standard errors to account for the GP practice based sampling in the fixed effect analysis and have now added this detail to the methods section as well. It is also worth noting that the survey is designed to give reliable estimates of patient experience for individual practices and so it is generally very well powered for analyses such as these where inequalities are considered.

8. On page 15: 'trans and non-binary respondents reported experiences of communication about 15 percentage points poorer'. It is not clear which results this statement relates to, as no differences are

as large as 15 pp. Might this be an error (if so please correct) or have been derived from unreported analyses (if so please add details)?

Many thanks for picking up this error. We have rechecked the analysis and have now corrected the wording in the results section to reflect the results in the table 3. This section now reads “After adjustment, trans and non-binary respondents were about half as likely to report a positive experience than all other survey respondents.”

Minor items:

Methods:

9. Please reference the STATA software where mentioned:

<https://www.stata.com/support/faqs/resources/citing-software-documentation-faqs/>

We have added this detail to the methods section.

Results:

10. Page 12 paragraph 2; “MH” is not defined/expanded.

We now write “mental health” in full where we previously used this shortening. Many thanks for picking this up.

Discussion:

11. In the 4th discussion paragraph, when possible reasons for differences in age-related patterns are given – “These patterns are likely the result of complex interactions between minority stress, behavioural risk factors, and biological effects of exogenous hormones, as well as sexual orientation and socioeconomic status.” – perhaps more open wording might better convey the idea that the exact causes are not fully known by adding ‘among others’ to the end. Other likely causes could be due to increased GP engagement meaning increased chance of diagnosis.

We have clarified this section in fourth paragraph of the discussion to include “among other possible mechanisms” and include healthcare access in our list of possible drivers.

12. Please expand on the potential impact of misclassification (given as a possible explanation for Alzheimer’s results) on study results/findings.

We have expanded this section, which now reads “This is one possible explanation for the association seen with dementia, where numbers are low, and inaccuracies (associated with cognitive decline) in inaccurately endorsing a trans and non-binary response may be higher.”

Tables and figures:

13. Figure 1:

a. Please include a legend for the line assignment.

This is included in the Figure heading. "Black line - all trans and/or non-binary respondents; Grey - all other survey respondents"

b. The continuous plot of a discrete scale could be misleading, especially with the jump to the 75+ cat (that may not be an even step-size). It may be less misleading to have a continuous x-axis with point estimates placed at the median age for that group. Alternatively, presenting disconnected points with error bars would be suitable.

This is a good point, and we did try adding error bars to earlier versions of figure 1. This just resulted in figures which looked really messy. The 95%CI for the model estimates (based on categorical age groups) which underlie these figures are presented in Appendix Table 4 and we now signpost these from the text "Adjusted percentages stratified by age are presented in Figure 1 and Appendix table 4 for model estimates, including confidence intervals." And so we are certainly not being misleading with the figure as the full set of estimates are available.

Unfortunately we can't include the median age for each group as age was self-reported in these categories, rather than the categories being created from linear data.

14. Table 3: Suggest renaming "overall experience" to 'Positive Overall Experience' to make clear that the variable has been dichotomised (otherwise it is not clearly binary).

We have made this change in Table 3.

15. Table 4: "Adjusted % endorsing the most positive response option / Adjusted % endorsing the most negative response option" – should these be 'weighted percentage'?

No, this is an adjusted percentage (adjusted for age, deprivation and ethnicity). We did not include weights in the estimation of these adjusted percentages and have included more information about the weighting approach in the methods section

Reviewer: 3

Thank you for your attention to this important topic.

In the introductory paragraph (page 5 of 35), focus is placed on the primary care provider's responsibility towards addressing long-term health conditions. The care of trans and non-binary persons, which is the true focus of this article, is only introduced in the final sentence of this paragraph. Consider revising this first paragraph to better present the topic and place emphasis on the population of focus rather than the setting. Additionally, noting care of trans and non-binary individuals after discussion of "long-term health conditions" may lead a reader to infer that gender identity is a "health condition" by itself.

Thanks for this helpful comment. We have deleted the middle section of the first paragraph of the introduction so that it is now more streamlined and clear that this is an article about trans and non-binary adults in primary care.

Page 5, line 19 - "gender reassignment" is not frequently used in US terminology. Unclear if this commonplace if UK scientific literature.

We have clarified the meaning of gender reassignment within the context of protected characteristics in the second paragraph of the introduction.

In the discussion section (pages 16-19), results were well presented and explored. Suggest further clarification that the study findings do not suggest a causative relationship between trans and non-binary gender identities and the various neuropsychological conditions where greatest disparities existed. While this may seem unnecessary given the clear statistics and discussion provided, in the current sociopolitical climate, tremendous care should be taken to protect vulnerable populations from harmful attempts at misrepresenting data.

We have clarified in the discussion that "Longitudinal population-based data collections are needed to understand causal relationships and properly disentangle the impacts of coming out at different ages on our understanding of these health impacts."

VERSION 2 – REVIEW

REVIEWER	Teece, Lucy University of Leicester, Department of Health Sciences
REVIEW RETURNED	13-Jan-2023

GENERAL COMMENTS	Thank you for addressing all of our comments, we are happy to recommend the article the article be accepted.
--

REVIEWER	Gianelis, Kristin Frontier Nursing University
REVIEW RETURNED	24-Jan-2023

GENERAL COMMENTS	Recommend for publication. Thank you for your excellent work.
---